# Identification of Medicinal Compounds of Fagopyri Dibotryis Rhizome from Different Origins and Its Varieties Using UPLC-MS/MS-Based Metabolomics

**DOI:** 10.3390/metabo12090790

**Published:** 2022-08-25

**Authors:** Chengcai Zhang, Yang Jiang, Changzheng Liu, Linyuan Shi, Jintong Li, Yan Zeng, Lanping Guo, Sheng Wang

**Affiliations:** 1State Key Laboratory Breeding Base of Dao-Di Herbs, National Resource Center for Chinese Materia Medica, China Academy of Chinese Medical Sciences, Beijing 100700, China; 2Dexing Research and Training Center of Chinese Medical Sciences, Dexing 334220, China; 3China National Traditional Chinese Medicine, Co., Ltd., Beijing 100191, China

**Keywords:** *Fagopyrum dibotrys*, UPLC-MS/MS, phenolic metabolites, PCA, medicinal differences

## Abstract

*Fagopyrum dibotrys*, being native to southwest China, is widely distributed in Yunnan, Guizhou Provinces and Chongqing City. However, the quality of medicinal materials growing in different origins varies greatly, and cannot meet the market demand for high-quality *F. dibotrys*. In this study, 648 metabolites were identified, and phenolic compounds of *F. dibotrys* from different origins were clearly separated by principal component analysis (PCA). Our results suggested that the medicinal differences of *F. dibotrys* from different origins can be elucidated via the variations in the abundance of the phenolic and flavonoid compounds. We found that the epicatechin, total flavonoids and total tannin content in Yunnan Qujing (YQ) and Yunnan Kunming (YK) were higher than those in Chongqing Shizhu (CS), Chongqing Fuling (CF) and Guizhou Bijie (GB), suggesting that Yunnan Province can be considered as one of the areas that produce high-quality medicinal materials. Additionally, 1,6-di-*O*-galloyl-β-*D*-glucose, 2,3-di-*O*-galloyl-*D*-glucose and gallic acid could be used as ideal marker compounds for the quality control of *F. dibotrys* from different origins caused by metabolites, and the *F. dibotrys* planted in Yunnan Province is well worth exploiting.

## 1. Introduction

*Fagopyrum dibotrys*, also known as golden buckwheat in China, is a perennial herb that belongs the *Polygonaceae* family of the genus Fagopyrum [1]. As an important traditional Chinese medicine (TCM) with high medicinal value, *Fagopyri Dibotryis* Rhizome (FDR) is often used for treating snakebite, traumatic injuries, lung diseases, inflammation, dysmenorrhea, lumbago, rheumatism, cancer, etc.; in particular, it has a reputation as being effective in treating lung cancer [2,3]. A considerable number of pharmacological experiments, both in vivo and in vitro, have validated that FDR possesses anti-diabetic, antitumor, anti-inflammatory, antioxidant, hepatoprotective, etc., activities [4]. Wild *F. dibotrys* is native to southwest China, and is widely distributed in the following regions: south of the Yellow River, the middle and lower reaches of the Yangtze River and the Pearl River Basin (21~32° N; 97~121° E). *F. dibotrys* grown in different ecological environments result in great changes in rhizome morphology and active-ingredient content. Meanwhile, the differences of origin of FDRs in the market has brought uneven quality, which seriously affects the clinical efficacy. In China, only certain FDRs that grow in specific geographic regions (Yunnan) can be used in Weimaining Jiao Nang, a Chinese medicine specifically designed for the treatment of lung cancer. However, the exact compounds underlying this origin variation of FDR are unknown.

Several grounds of bioactive chemical constituents, namely flavonoids, phenolics, tannins, cyclitols and triterpenoids, have been isolated from *F. dibotrys* [5,6]. Among these, phenolic compounds and flavonoids were generally acknowledged to be the main active components of FDR, including hyperin, epicatechin, protocatechuic acid, 3-*O*-methylquercetin, kaempferol, catechin, luteolin, rutin, procyanidin C1, etc. [7,8]. Flavonoids in FDR exhibited remarkable biological functions in antioxidant, antidiabetic, anti-inflammatory and antihypertensive aspects [5,6]. The condensed tannins isolated from FDR showed excellent anti-tumor and antioxidant effects [9,10]. The content of epicatechin, a flavanol, is one of the key indicators for quality assessment of FDR, in which the Chinese Pharmacopoeia (2020) stipulates that epicatechin (C15H14O6) in FDR shall not be less than 0.030% [11]. Thus, a study of the detailed profile of metabolites in FDR would be important for comprehensively evaluating its medicinal value, explicating compounds underlying this original variation and understanding its related efficacy. Metabolomics as a technology for the simultaneous qualitative and quantitative analyses of all the metabolites in living beings is introduced; this is widely employed in plant [12], medicine [13], food [14], microorganism [15] and other research fields because of its high efficiency, comprehensiveness and accuracy. Metabolomics is focused on the quantification and identification of the changes in metabolites due to environmental, planting or genetic factors. Yajun et al. analyzed the differences of metabolic components of *Lycii fructus* in different production areas through widely targeted metabolomics and found that climate factors have a great impact on the nutrition and medicinal quality of fruits of *Lycium barbarum* from different planting areas [16]. Jing et al. analyzed the differences of flavonoids in buckwheat leaves by metabolomics and found that there were great differences [17]. To date, a comprehensive and systematic study on the differences of chemical components of FDR from different habitats has not been published. In the present research, ultra-performance liquid chromatography mass spectrometry (UPLC-MS/MS), hierarchical clustering analyses (HCAs), PCA analyses and orthogonal projections to latent structure discriminant analyses (OPLS-DA) were carried out to study the differences of metabolites in FDR from different growing areas and find their common differential metabolites. Our work provides valuable data for assessing the medicinal value to guide the standardized planting of *F. dibotrys*.

## 2. Materials and Methods

### 2.1. Plant Materials

Fifteen rhizomes of two years *Fagopyrum dibotrys* were collected in the autumn (11–25 November 2020) from 5 different producing areas: Shizhu in Chongqing (CS), Fuling in Chongqing (CF), Qujing in Yunnan (YQ), Kunming in Yunnan (YK) and Bijie in Guizhou (GB). We chose 3 sites as replicates for each producing area, and 10 individuals were collected in each site as one sample for analysis after being dried at 40 °C and powdered. Detailed information of samples is displayed in Appendix A.

### 2.2. Quantification of Epicatechin, Total Flavonoids and Total Tannins

The contents of epicatechin in FDRs from five growing areas were determined via high performance liquid chromatography (HPLC). In detail, sample processing was as follows: dried sample was ground into powder, then filtered through a sieve with 250 ± 9.9 μm diameter pore size (a Size No. 4 sieve defined by the Chinese Pharmacopoeia (2020) [11]. Approximately 4.0000 g of precisely measured sample powder was placed into a conical flask; then, 100 mL of precisely measured 10% (*v*:*v*) acetonitrile was carefully mixed with the sample powder in the flask. The flask was then sealed with the plug and its total weight was measured as the initial weight. The mixture was then allowed to stand for one hour and subsequently boiled for one hour before being allowed to stand again until it cooled down to room temperature. The weight was measured again and 10% (V:V) acetonitrile was added into the sample mixture to the initial weight. The mixed sample crude extract was filtered twice with a 0.2 μm PVDF filter to acquire the final sample extract to be subjected to HPLC analyses. A 4 μL amount of the filtered sample exact was injected into an HPLC system (Alliance 2695, Waters Corp., Milford, CN, USA) with a Symmetry C18 column (250 mm × 4.6 mm, 5 μm particle size); sample separation was performed with a gradient elution (V:V) program using acetonitrile (chromatographic purity, Aladdin) and X M phosphate buffer saline (pH = 3.0) as the mobile phase at a constant column temperature of 50 °C following the gradient below: 8% acetonitrile for 25 min; 8% to 9% acetonitrile for 20 min; 9% to 10% acetonitrile for 15 min; 10% to 8% acetonitrile for 5 min. Detection of separated sample was performed with a photo-diode array detector under 280 nm.

As for total flavonoids and total tannins, the content of total flavonoids was indicated by the absorbance value of the extract solution of FDR at 282 nm. Total tannin content was determined with insoluble polyvinyl-polypyrrolidone (PVPP), which binds tannins [17]. Briefly, 1 mL of extract (1 mg/mL) was mixed with 100 mg of PVPP, vortexed, kept for 15 min at 4 °C and then centrifuged for 10 min at 3000 rpm.

### 2.3. Sample Preparation for Metabolomic Analysis

Fifteen FDRs from five different producing areas were handled via vacuum freeze-drying (Scientz-100F, ANPEL, Shanghai, China). Then, the freeze-dried FDR sample was ground into fine powder via a mixer mill (MM 400, Retsch), and the setting parameters of the grinding instrument were: 30 Hz, 1.5 min. Then, 100 mg of lyophilized powder of FDR was weighed and dissolved with 1.2 mL 70% methanol solution. After that, the extract solution of FDR was vortexed for 30 s every 30 min, repeatedly 6 times in total, and then placed at 4 °C inside a refrigerator overnight. After centrifugation at 12,000× *g* rpm for 10 min, supernatant was collected as the crude extract. The extract samples were then filtrated by SCAA-104 with a 0.22 μm pore size (ANPEL, Shanghai, China) before UPLC-MS/MS analysis. The conditions and methods of UPLC-MS/MS can be found in Shang, X et al. [18] (details provided in Appendix A).

### 2.4. Multivariate Statistical Analysis

The qualitative and quantitative analysis of 15 FDRs from five different producing areas was performed according to the method of Zhang, J. et al. [19] and Fraga, C et al. [20] (details provided in Appendix A). Metabolite data of *F. dibotrys* from 15 samples were used for unsupervised PCA, HCA and supervised multiple regression OPLA-DA using the Met ware Cloud. The results of HCA of the metabolites and samples were presented as heatmaps with dendrograms. The differentially accumulated metabolites (DAMs) were identified based on the fold-change (Log2FC ≥ 2 or ≤ 0.5) and variable importance in project scores (VIP ≥ 1), and the VIP values were extracted from OPLS-DA results. The identified 187 phenolic metabolites were annotated according to the Kyoto Encyclopedia of Genes and Genomes (KEGG) database (http://www.kegg.jp/kegg/compound/ (accessed on 24 June 2022)) with a *p*-value < 0.01, and the pathways with differentially phenolic metabolites mapped were then fed into metabolite sets enrichment analysis (MSEA).

## 3. Results

### 3.1. Determination of Epicatechin, Total Flavonoids and Total Tannin in FDR from Different Producing Areas

In this study, FDRs from five different producing areas—Shizhu in Chongqing (CS), Fuling in Chongqing (CF), Qujing in Yunnan (YQ), Kunming in Yunnan (YK) and Bijie in Guizhou (GB)—were collected for analysis. Considered active components, the content of epicatechin, total flavonoids and total tannin from different producing areas were investigated. As shown in Figure 1, contents of epicatechin in YQ and YK were significantly higher (*p* < 0.05) than in CF and GB. Among them, contents of epicatechin of YK is the highest (0.113%); this is considered to be the best quality according to the Chinese Pharmacopoeia; GB has the lowest epicatechin content of 0.038%. We used Abs at 280 nm to indicate the relative amount of total flavonoids of FDR samples; the flavonoid content of YK and YQ are significantly higher (*p* < 0.05) than that of GB, CS and CF, whereas YK is slightly higher than that of YQ, but the difference was not significant. Similarly, the total tannin content of YQ was significantly higher than that of GB, CS and CF. Nevertheless, there was only a tendency higher than that of YK, which is also slightly rich in content of tannins compared to CS, CF and GB. In addition, the cross-sectional colors of YK, CS and GB rhizomes were reddish brown, light red and yellow white, which were consistent with the change trend of total flavonoid content (Appendix A). These results revealed a clear association of chemical composition with distinct FDRs in different producing areas, which provides representative materials for identification of differentially accumulating metabolites related to origin.

### 3.2. Metabolic Profiling of FDR

To better understand the metabolic differences from different origins of FDR, the metabolic profiles of FDRs from five producing areas were constructed based on UPLC-MS/MS analysis. As a result, 648 metabolites (Appendix A) were detected, including 120 flavonoids, 7 tannins, 9 quinones, 131 phenolic acids, 50 amino acids (and derivatives), 30 nucleotides (and derivatives), 21 lignans and coumarins, 49 alkaloids, 20 terpenoids, 48 organic acids, 94 lipids and 69 additional compounds that did not fit into these 11 main classes (Figure 2a). The results show that secondary metabolites accounted for a relatively high proportion in metabolic profiles for FDRs of different origins. The cluster heatmap of the metabolites effectively demonstrated similarity among the compounds of biological repeats, and differences in the components among the FDR samples from different origins (Figure 2b).

### 3.3. Differential Analysis of Phenolic Compounds Based on PCA

Natural phenolic compounds have an outstanding role in preventing and treating cancer. Thus, five phenolic compounds (flavonoid, phenolic acid, quinones, tannins, and lignans and coumarins) likely contributing to the medicinal quality were subjected to PCA. The PCA results showed that the first two principal components of FDRs of different origins could explain 54.9% of variance contribution value; PC1 and PC2 were 43.1% and 11.8%, respectively (Figure 3). In the PCA score plot, CS, CF, GB and YK, YQ and quality control (QC) were clearly separated, and the biological repeats of QC were closely clustered, showing the repeatability of the experiments and reliability of our data. The results of PCA suggested that five separated clusters were associated with different metabolite profiles of FDR from different producing areas, and CS, CF and GB belong to the non-Yunnan (NYN) group and YK and YQ belong to the Yunnan (YN) group.

### 3.4. Differential Analysis of Phenolic Compounds Based on OPLS-DA

To further illustrate the differences of 187 phenolic compounds, OPLS-DA was performed in the comparison group of FDRs from different origins. The OPLS-DA analysis is a multivariate statistical analysis method with supervised pattern recognition, which can effectively eliminate the effects unrelated to the study. Subsequently, high predictability (Q^2^) of the OPLS-DA models was observed in the 10 comparison groups. The results of Figure 4 showed that the OPLS-DA models of all comparison groups were reliable and stable (R^2^X > 0.8, Q^2^ > 0.9), which could be used to screen for even more differential phenolic metabolites (DPMs).

### 3.5. Differential Phenolic Analysis

We further performed DPM screening among all 187 phenolic compounds based on the fold-change (Log_2_FC ≥ 2 or ≤ 0.5) and variable importance in project (VIP ≥ 1) scores. The DPM results are shown as upset plots in detail (Figure 5 and Appendix A). Among these, there were five DPMs (four down-regulated and one up-regulated) in YQ compared to YK, six between CS and CF (four down-regulated in CF), seven between CF and GB (five down-regulated in GB), 15 between CS and GB (12 down-regulated in GB), 20 between CS and YK (eight down-regulated in CS), 36 between CS and YQ (19 down-regulated in CS), 24 between CF and YK (14 down-regulated in CF), 40 between CF and YQ (27 down-regulated in CF), 26 between GB and YK (14 down-regulated in GB) and 50 between GB and YQ (36 down-regulated in GB). The number of DPMs was the highest between the samples collected in GB and YQ (50 DPMs in total); relatively high numbers of DPMs were also identified between CF and YQ (40 DPMs), and CS and YQ (36 DPMs). In addition, compared with the YN group (YQ, YK), most of the identified DPMs of the NYN group (GB, CF) were down-regulated, whereas CS vs. YK was not. These results showed evident differences in phenolic metabolites between samples collected in the YN and NYN groups. In contrast, only five DPMs were identified between the samples from YK and YQ, reflecting comparatively slight difference in phenolic metabolites among the samples from habitats within Yunnan Province.

The DPMs were classified among different comparison groups in an upset plot (Figure 5). Three common DSMs were shown amongst comparison groups CS vs. YQ, CF vs. YQ, GB vs. YQ, CS vs. YK, CF vs. YK and GB vs. YK, of which 1,6-di-*O*-galloyl-β-*D*-glucose, 2,3-di-*O*-galloyl-*D*-glucose and gallic acid were classified as phenolic acid. When comparing the metabolite ion intensity between FDRs from the YN group (YQ, YK) and the NYN group (GB, CF, and CS), we observed a significant increase in the content of 2,3-di-*O*-galloyl-*D*-glucose, 1,6-di-*O*-galloyl-β-*D*-glucose and gallic acid in FDRs from the YN group (YQ, YK) compared to those from the NYN group (GB, CF and CS) (Figure 6). The results showed that the three DPMs could be used as ideal chemical markers for the quality control of *F. dibotrys* from different origins.

In addition, nine DPMs (isorhamnetin-3-*O*-glucoside, isorhamnetin-7-*O*-glucoside (brassicin), nepetin-7-*O*-glucoside, phloretin, apigenin-3-O-glucoside, luteolin-8-C-glucoside (orientin), limocitrin-7-*O*-glucoside, fraxetin-7,8-di-*O*-glucoside and piperitol) can be observed only among comparison groups of CS vs. YQ, CF vs. YQ and GB vs. YQ, and one DPM (2-methoxybenzaldehyde) was observed only among comparison groups CS vs. YK, CF vs. YK and GB vs. YK.

A pathway enrichment analysis of KEGG of 187 phenolic compounds (Appendix A) between the NYN and YN groups was also performed. The enrichment analysis results of DPMs in NYN vs. YN were mainly enriched in ‘phenylpropanoid biosynthesis’ (8), ‘the pathway of flavone and flavonol biosynthesis’ (11), ‘flavonol biosynthesis’ (17), ‘metabolic pathways’ (25) and ‘biosynthesis of secondary metabolites’ (24) (Figure 7 and Appendix A). These results show that environmental and climate factors have a strong impact on the biosynthesis of the secondary metabolites of FDRs. The flavonoid profiles of the *F. dibotrys* samples from the habitats either inside or outside Yunnan province each had significance and distinct variance.

### 3.6. Metabolite Differences among NYN Group and YN Group

We focused on classes of metabolites likely to be major contributors to the medicinal and nutritional qualities among FDRs of the NYN and YN groups. Based on Log2FC (fold change) and VIP values, we screened out 44 up-regulated substances in the YN group compared to the NYN group (Figure 8 and Appendix A). In detail, 41 phenolic metabolites presented comprise 12 phenolic acids, 24 flavonoids, two quinones and three lignans and coumarins. Of these, the majority of flavonoid phenolic acids were significantly greater in the YN group than in the NYN group (Log 2 FC ≥ 2), whereas the majority of quinones and lignans and coumarins exhibited differences (1 ≤ Log 2 FC ≤ 2), suggesting that these phenolic metabolites are involved in the medicinal differences between the YN group and the NYN group. Furthermore, three primary differential metabolites comprising *L*-cyclopentylglycine, *N*-methyl-trans-4-hydroxy-*L*-proline and *L*-asparagine were also identified; these are involved in the nutritional differences between the YN group and the NYN group. These results showed that among the DAMs between the *F. dibotrys* from Yunnan and non-Yunnan habitats, the majority were up-regulated in the secondary metabolites of FDRs. Therefore, it could be speculated that compared with primary metabolites, the biosynthesis of the secondary metabolites was more responsive to environmental factors. This might be one of the reasons why herbs from different habitats would exert distinct therapeutical effects; meanwhile, it places further importance on the geo-authenticity of traditional Chinese medicinal plants.

## 4. Discussion

Southwest China is an internationally recognized center of buckwheat origin, among which the wild *F. dibotrys* resources in Yunnan, Guizhou and Chongqing are the most concentrated [22,23,24,25]. *F. dibotrys rhizomes* have been used to develop different health products due to their abundant bioactive phenolic compounds. However, most FDRs circulating in the market of medicinal materials have different origins, and their quality is uneven. The quality difference of FDRs grown in different habitats will eventually be reflected in the terminal metabolites of the metabolic pathway. Thus, this study aimed to provide evidence of metabolomics-based medicinal compound identification of FDR from different origins.

### 4.1. Metabolic Profiling of FDR from Different Origins

To comprehensively analyze the medicinal values of FDRs in different *F. dibotrys*-producing areas, the results of metabolome analysis showed that 648 metabolites were identified, including 120 flavonoids, 7 tannins, 9 quinones and 131 phenolic acids, 50 amino acids and derivatives, 30 nucleotides and derivatives, 21 lignans and coumarins, 49 alkaloids, 20 terpenoids, 48 organic acids and 94 lipids. Phenolic compounds, flavonoids (catechin and epicatechin), γ-aminobutyric acid and terpenoids have been reported in FDRs [26]. In our study, we found that alkaloids, organic acids and lipids were also ranked in FDRs; the metabolic profilings of *F. dibotrys* were further extended. In so far as we know, our current study is the first metabolome report to differentiate varieties of *F. dibotrys* from different origins. The establishment of the metabolite profiles of *F. dibotrys* from different habitats is important for elucidating the differential therapeutical effects and clinical values of FDR. Meanwhile, it is also important for further understanding the complex effect and underlying mechanisms of traditional Chinese herbs. The results of this study are particularly meaningful for evaluating the quality of *F. dibotrys* resources, developing standardized *F. dibotrys* farming systems and establishing a quality control system for geo-authentic *F. dibotrys* products. These results show that the synthesis of metabolites of *F. dibotrys* is induced by the environment under different growth areas, which is consistent with the detection results of different metabolic substances in different ecotypes for *Cistanche deserticola*, indicating that the production environment is the main factor affecting the accumulation of metabolites of medicinal plants [27,28,29].

### 4.2. Differential Phenolic Metabolite Analysis of FDR from Different Origins

Fifteen FDR samples obtained from five *F. dibotrys*-producing areas were collected to identify the differences in metabolite extracts. PCA results show that the metabolites of FDRs from different origins are obviously different, and could be divided into two major subgroups. The rhizomes of YK and YQ show higher total flavonoids, particularly epicatechin, being a major bioactive element in Chinese pharmacopoeia. In addition, the total tannin content of FDRs in YQ was the highest, followed by YK, CS, CF and GB. These findings revealed five distinct groups associated with distinct metabolite profiles of FDR from different producing areas, and CS, CF and GB belong to the non-Yunnan (NYN) group and YK and YQ belong to the Yunnan (YN) group. The number of DPMs was the highest between the samples collected in GB and YQ (50 DPMs in total); relatively high numbers of DPMs were also identified between CF and YQ (40 DPMs), and CS and YQ (36 DPMs). Moreover, the YN group (YQ, YK) had more DPMs than the NYN group (CS, CF and GB), the up-regulated metabolites of which account for the majority of all DPMs (Figure 5). In contrast, only five DPMs (four down-regulated in YQ and one up-regulated) were identified between the samples from YK and YQ, reflecting comparatively slight difference in phenolic metabolites among the samples from habitats within Yunnan Province. These results showed evident differences in phenolic metabolites between samples collected in the YN and NYN groups.

A number of phenolic compounds, namely phenolic acids, tannins, flavonoids, quinones, coumarins, lignans and so on, were extracted from multiple medicinal herbs or dietary plants, and these compounds were proven to have important roles in preventing and treating lung cancers [30]. Among the above phenolic compounds, flavonoids were acknowledged as the most functionally valuable components; they were reported to be beneficial to health, for example, because they enhance vascular toughness, are good for treating gastrointestinal dysfunction, reduce blood sugar and blood fat and are good for treating inflammation; also, they can improve immunity and promote tumor inhibition [31,32]. The present results suggested that the level of flavonoid compounds increased in YN group compared with NYN group. In addition, our results show that there is a higher content of some flavonoids and phenolic acids in the YN group, such as isorhamnetin 3-*O*-glucoside, isorhamnetin-7-*O*-glucoside (brassicin), apigenin-3-*O*-glucoside, nepetin-7-*O*-glucoside, phloretin, piperitol, fraxetin-7,8-di-*O*-glucoside, syringetin-7-*O*-glucoside, syringetin, limocitrin-3-*O*-glucoside and 2-methoxybenzaldehyde (Figure 8), which possess anti-inflammatory, anti-cancer, anti-oxidant, anti-bacterial and anti-diabetic activities, suggesting that FDRs from Yunnan province have a better medicinal value and are worth exploiting. Our findings confirmed that the metabolites with the greatest variation in the FDRs of five producing places were phenolic and flavonoids. Based on these results, the phenolic and flavonoid compounds could be selected to assess the quality of FDRs from different origins. The enrichment analysis results indicate that the DPMs were significantly enriched in metabolic pathways, biosynthesis of secondary metabolites and flavone and flavonol biosynthesis. That is the reason why we think that YQ and YK were divided into the high medicinal value (YN) group, and CS, CF and GB were classed as the low medicinal value (NYN) group.

### 4.3. The Common DPMs of FDR from Different Origins

Gallic acid, a predominant polyphenol, has the simplest molecular structure among all the natural polyphenols. In recent years, remarkable anti-tumor effects of gallic acid on cancers, including pancreatic cancer, lung cancer, prostatic cancer and skin cancer, have been confirmed through in vitro and in vivo approaches [33,34,35]. Gallic acid was reported to have significant anti-tumor effects: gallic acid and its derivatives can inhibit the proliferation of tumor cells, induce the apoptosis of tumor cells and suppress tumor metastasis; meanwhile, gallic acid can exert these anti-tumor functions with high selectivity, specificity and low damage concerning the healthy cells. The major traditional Chinese medicinal plants that are rich in gallic acid include *Galla chinensis* [36], *Cornus officinalis* [37], Pomegranate [38], *Rheum palmatum* L. [39] and *Moutan cortex* [40]. 1,6-di-*O*-galloyl-β-*D*-glucose and 2,3-di-*O*-galloyl-*D*-glucose have been isolated from the leaves of *Castanopsis fordii* Hance and the rhizomes of rhubarb, which had a strong effect reducing blood pressure [41,42,43]. Shao et al., studying the phenolic acid derivatives from the rhizome of *Fagopyrum cymosum*, showed the pharmacological material basis of FDRs [44]. At present, approximately 20 phenolic derivatives in *F. dibotrys* have been detected, including gallic acid, syringic acid, 3,4-dihydroxy benzoic acid, succinic acid, etc. [45]. The results of common DPM analysis of CS vs. YQ, CF vs. YQ, GB vs. YQ, CS vs. YK, CF vs. YK and GB vs. YK suggested that 2,3-di-*O*-galloyl-*D*-glucose, 1,6-di-*O*-galloyl-β-*D*-glucose and gallic acid could be used as ideal chemical markers for the quality control of FDR from different origins. Moreover, the three common DPMs were higher in the YN group (YQ, YK) than the NYN group (GB, CF and CS).

In addition to the medicinal values, three primary differential metabolites contributing to its nutritional value were also identified. In detail, the contents of *L*-cyclopentylglycine, *N*-methyl-trans-4-hydroxy-l-proline and *L*-asparagine were significant higher in the YN group than in the NYN group. Previous studies reported that *F. dibotrys* contained 18 amino acids, multiple vitamins and essential inorganic substances, which proved its important nutritional value as a silage additive and health food [7,46]. Hence, *F. dibotrys* is a herb that is further worth exploiting.

### 4.4. The ‘Good Traits’ of the High-Quality Fagopyri Dibotryis

The ‘good traits’ of traditional Chinese medicine (TCM) could be used as one of the identification criteria of the high-quality herbs, such as ‘Yunjin huawen’ of Heshouwu (in Chinese), which is a pattern formed by anomalous vascular bundles and can be observed on the cross-section of the root tuber of *Fallopia multiflora* (Thunb.) Harald [47]. Hence, the cross-sectional colors of YK, CS and GB rhizomes were reddish-brown, light red and yellow white, which was consistent with the change trend of total flavonoids and total tannin content. This illustrates that the reddish-brown section of FDRs may contribute to its high quality as a ‘good trait’. In addition, the results showed that absorbance values and total tannin content of FDRs were also positively correlated with its section color. Anthocyanins belong to the flavonoid family that widely exists among the leaves, roots and flowers of most fruits and some medicinal plants [48,49]. Zhang et al. found that delphinidin 3-*O*-glucoside, malvidin 3-*O*-glucoside and delphinidin may be the key anthocyanins conferring the red pigmentation of jujube peel over the fruit ripening periods [50]. The three anthocyanins, therefore, could be regarded as a red coloration of plant organs. In our current findings, we found that procyanidin B3, procyanidin B1, procyanidin B2 and procyanidin B4 of FDRs from different origins were identified; their relative contents were significantly different. Considering the possible coloring mechanism, these four metabolites may lead to the changes from reddish-brown to yellow white in FDRs of YK, CS and GB.

## 5. Conclusions

Identification of medicinal compounds and metabolite profiling analyses of *Fagopyrum dibotrys* rhizomes (FDR) indicated that the medicinal differences of FDR from different origins can be elucidated via the variations in the abundance of phenolic and flavonoids. The metabolic profiles of *F. dibotrys* from different origins were significantly enriched. We found that the epicatechin, total flavonoids and total tannin content in Qujing in Yunnan (YQ) and Kunming in Yunnan (YK) were higher than those in Shizhu in Chongqing (CS), Fuling in Chongqing (CF) and Bijie in Guizhou (GB), consistent with the traditional view that Yunnan Province can be considered as one of the producing areas of high-quality medicinal materials. After that, a medicinal-difference analysis of *F. dibotrys* from different origins was successfully performed. In detail, phenolic acids and flavonoids, such as isorhamnetin-7-*O*-glucoside (brassicin), isorhamnetin 3-*O*-glucoside, apigenin-3-*O*-glucoside, nepetin-7-*O*-glucoside, phloretin, piperitol, fraxetin-7,8-di-*O*-glucoside, syringetin, 7-*O*-glucoside, syringetin, limocitrin-3-*O*-glucoside and 2-methoxybenzaldehyde were significantly higher in the YN group. Moreover, phenolic and flavonoids exhibited the greatest variation among the metabolic profiles of FDRs from different origins, which could be used to assess the quality of raw medicine. Additionally, 2,3-di-*O*-galloyl-*D*-glucose, gallic acid and 1,6-di-*O*-galloyl-β-*D*-glucose could be used as ideal marker compounds for the quality control of *F. dibotrys* from different origins.

## Figures and Tables

**Figure 1 metabolites-12-00790-f001:**
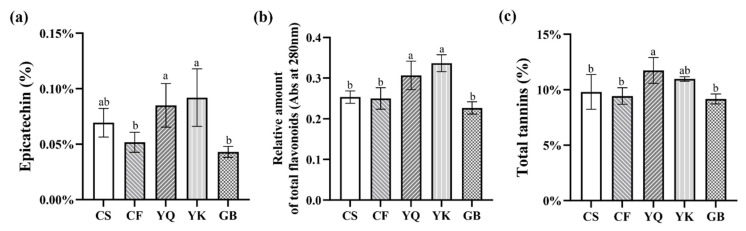
Content of active components in *Fagopyri Dibotryis* Rhizome (FDR) from different producing areas. (**a**) Epicatechin content of FDRs. (**b**) Total flavonoid content of FDRs. (**c**) Total tannin content of FDRs. The a, b, c above the bar chart indicates the difference significance; and the two comparison groups with no significant difference were marked with the same letter; The significance value was a > ab > b.

**Figure 2 metabolites-12-00790-f002:**
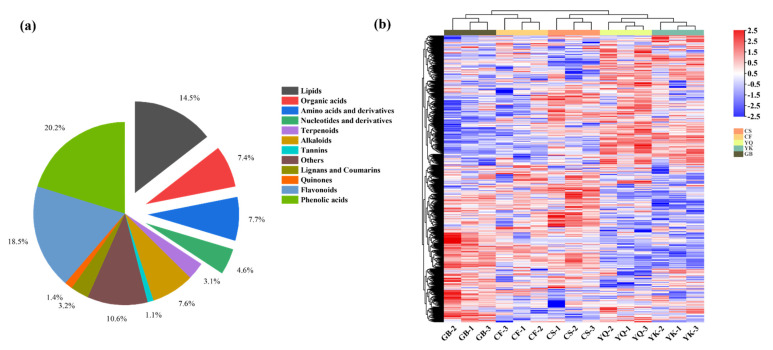
(**a**) Statistics of metabolites in all samples of FDR. (**b**) Heat map showing the relative variatons in the metabolic profiles of different origins of FDRs. (**a**) Pie chart visualization. The lipids, organic acids, amino acids and derivatives, and nucleotides and derivatives of the exploded pie diagram are primary metabolites. (**b**) Each sample from different origins is visualized with one column, and each metabolite of FDRs of different origins is represented by a raw element. Color gradient indicates the accumulation level of metabolite content (red: high abundance; blue: low abundance).

**Figure 3 metabolites-12-00790-f003:**
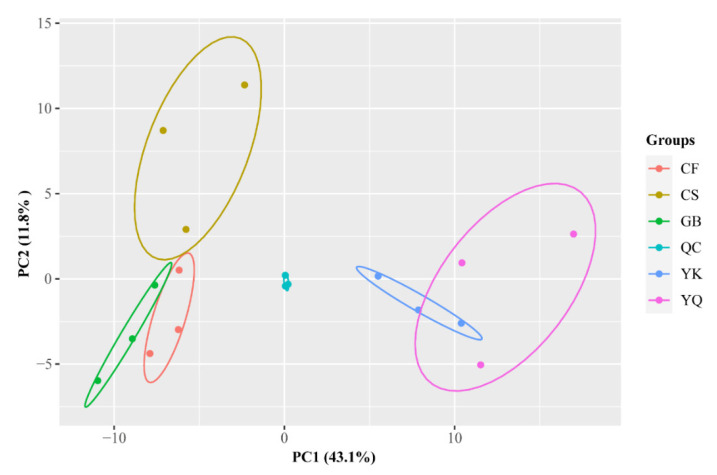
PCA of the relative differences in phenolic compounds of FDRs from different producing areas. Quality control (QC) samples mixed by equal volumes of all FDR rhizomes.

**Figure 4 metabolites-12-00790-f004:**
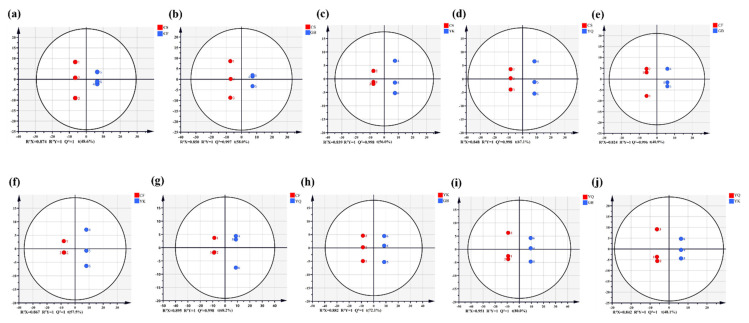
OPLS-DA score map of phenolic compounds of FDRs from different producing areas. (**a**–**j**) OPLS-DA model plots for the comparison group CS vs. CF, CS vs. GB, CS vs. YK, CS vs. YQ, CF vs. GB, CF vs. YK, CF vs. YQ, GB vs. YK, GB vs. YQ and YK vs. YQ, respectively.

**Figure 5 metabolites-12-00790-f005:**
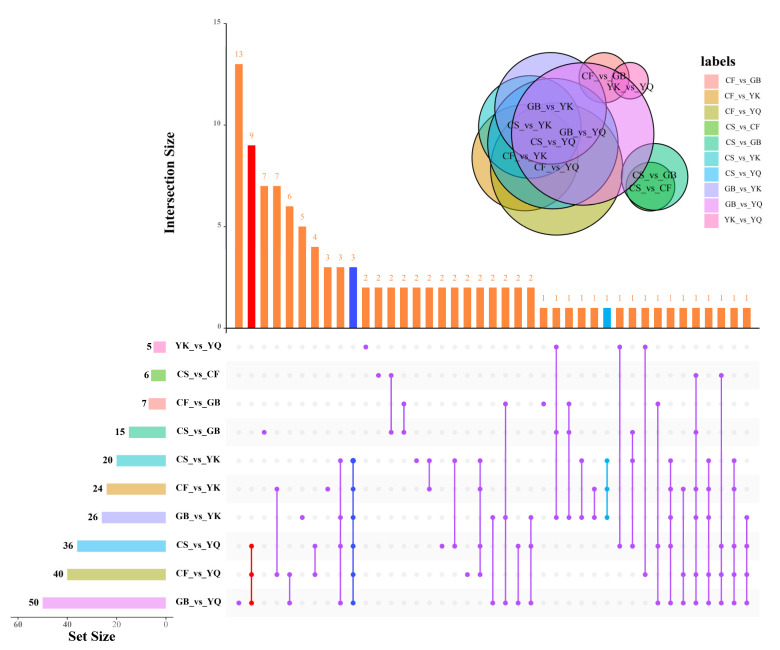
Upset plot of differentially accumulated phenolic metabolites of FDRs from different origins. Venn diagram and upset plot show sets and intersections of different comparison groups of FDRs. Refer to Tong, B. et al. [21] for legend description.

**Figure 6 metabolites-12-00790-f006:**
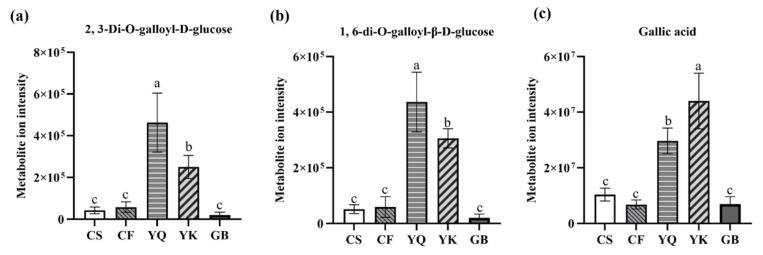
The total ion intensity of three classes of phenolic metabolites in FDRs from different producing areas. (**a**) 2, 3-di-*O*-galloyl-*D*-glucose, (**b**) 1,6-di-*O*-galloyl-β-*D*-glucose, (**c**) gallic acid. Bars represent the sum of ion intensity of all metabolites belonging to each class. The a, b, c above the bar chart indicates the difference significance; and the two comparison groups with no significant difference were marked with the same letter; The significance value was a > b > c.

**Figure 7 metabolites-12-00790-f007:**
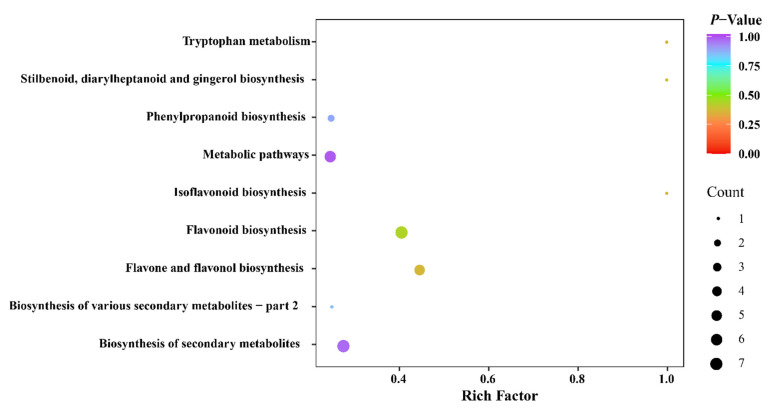
Metabolic pathway enrichment analysis for the NYN and YN groups differentially accumulated metabolites.

**Figure 8 metabolites-12-00790-f008:**
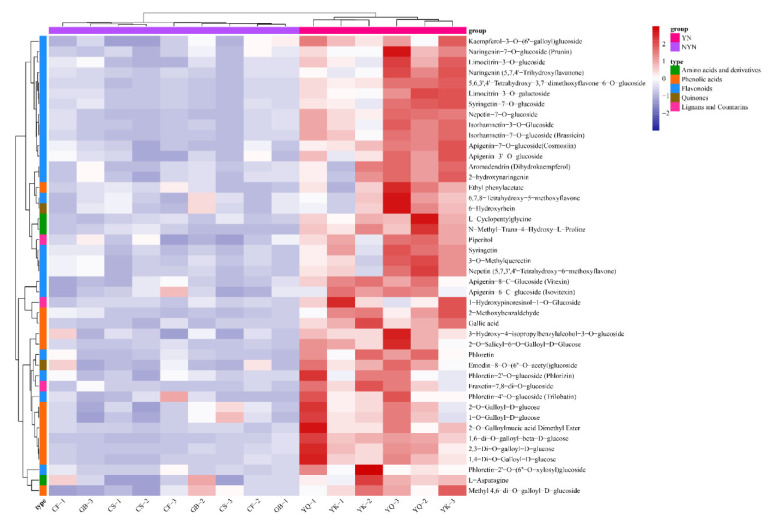
Heatmap of differential metabolites from FDR samples of NYN group and YN group. Red colors show high abundance, whereas low abundance is presented by blue.

## Data Availability

Not applicable.

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
