# Peer review of "Identification of Medicinal Compounds of Fagopyri Dibotryis Rhizome from Different Origins and Its Varieties Using UPLC-MS/MS-Based Metabolomics"

_metabolites, 2022, doi:10.3390/metabo12090790_

Round 1

Reviewer 1 Report

This manuscript written by Chengcai Zhang and coworkers described the Identification of medicinal compounds of Fagopyri dibotryis. 648 metabolites were identified and two specific compounds were determined as ideal chemical markers for quality control. I feel this paper cannot be published after minor revision.

The errors and comments are listed below:

1.      Lines 117-122, 2.3 sample preparation, please add a reference for your methods.

2.      In figure 2 (b), What is the black linkage in the very left? Name of metabolite? I can not see any word, just black lines mixed together.

3.      Figure 4 is blurred; please provide high-resolution pictures.

4.      Line 465, the “18” should be italic. All the volume numbers should be italic.

Author Response

Dear Editors,

The revised opinions of experts are attached as follows:

  • Please use the version of your manuscript found at the above link foryour revisions and revise your manuscript according to the referees’ comments and upload the revised file within 5 days.
  • The similarity in wording with the previous publication is high,rephrasing is needed with the revision. We have attached the similarity result for you in this email to use as a reference in rephrasing. Please reduce the similarity to be lower. The big similar section should be removed and rewritten in your own words. Consider a shorter text summarizing the main points and referring readers to the other publication for full details. For methods or theorems, make sure the original source is clearly cited and that readers would have the impression that the text is original.
  • Please check that all references are relevant to the contents of the
  • Any revisions made to the manuscript should be marked up using the“Track Changes” function if you are using MS Word/LaTeX, such that changes can be easily viewed by the editors and reviewers. 
  • Please provide a short cover letter detailing your changes for theeditors’ and referees’ approval.

The reply of manuscript revised is as follows:

  • The manuscript revision has been submitted as scheduled.
  • The manuscript repetition rate has been reduced to qualified, and the duplicate check report has been replied in the email. For methods or theorems, this part of the content has been mainly revised.
  • The manuscript references have been carefully checked.
  • Manuscript revision has used the "track changes" function, which had submitted on MDPI website.
  • Manuscript revision: the main revision parts are on the repetition rate and reference check. Formethods or theorems, “2.5. ESI-Q Trap-MS/MS” and “2.6. Qualitative and Quantitative Metabolite Analyses” were added to the manuscript revision.

Description of manuscript revision: we had revised some sentences where litaretures were cited to reduce similarity with past literature. Also, we checked the similarity index after our revision on the 'http://www.zaojiance.net/' website and made sure the similarity was reduced. The similarity index was 17% (Remove the parts of “2.1-2.3” of supplementary materials). However, in our 'Material and methods' section, the contents were impossible to revise to reduce the similarity, because our methods were not totally novel and original, but were tested by multiple previous studies, reliable and mature. hence, we moved some detailed contents of our methods to an additional Supplementary material in order to reduce the overall similarity of the manuscript. We hope it would be accepable."

Thank you and best regards.

Sincerely yours,

Chengcai Zhang

State Key Laboratory Breeding Base of Dao-di Herbs, National Resource Center for Chinese Materia Medica, China Academy of Chinese Medical Sciences, Beijing 100700, China.

Reviewer 2 Report

For Authors:

The manuscript was well written and it is very original and interesting. 

Three main remarks:
i) the English should be revised by a native English speaker;
ii) the plant material should be improved by adding more information (genotypes, varieties?);
iii) the conclusions should be written avoiding redundant information already discussed in the other sections and it should be focused only on the mail results in relation to the aims of the study.

A last suggestion: Results and Discussion sections may be integrated to a single section to avoid repetitions.

Author Response

(The authors gave the same response as above.)

Reviewer 3 Report

The work of Zhang et colleagues is well structured and well written. The treated topic adds relevant information on the presence of secondary metabolites of the investigated matrix.

Chemical analysis was conducted only to identify the non-volatile metabolomic profile. For what reason the authors have not also thought of an analysis by GC / MS of the extracts to identify any volatile metabolites?

In light of the above, I recommend removing the term "comprehensive" from the conclusions as the metabolomic analysis performed is partial.

In any case, I propose this work for publication after some minor revisions listed below:

1- How many times were UPLC-MS analyzes replicated for each sample?

 Are the values ​​expressed in Table S2 absolute or average values? Please add this information in the results paragraph.

2-Figure 8: improving the quality, the writings are faded and the names of the compounds are not easily legible.

3-I recommend to insert in the manuscript (not in SM), a table with the list of identified compounds expressing their relative quantities as percentages excluding those below 0.1%. This would facilitate the reader in understanding the main compounds characterizing the matrices.

Author Response

(The authors gave the same response as above.)
